# *Prosopis* Species—An Invasive Species and a Potential Source of Browse for Livestock in Semi-Arid Areas of South Africa

**Khuliso Emmanuel Ravhuhali** [1,2,*] , **Humbelani Silas Mudau** [1,2] , **Bethwell Moyo** [3] , **Onke Hawu** [1,2,*] and **Ntokozo Happy Msiza** [1,2]

1 Department of Animal Science, School of Agricultural Sciences, Faculty of Natural and Agricultural Sciences, North-West University, Mmabatho 2735, South Africa; mudausilas@gmail.com (H.S.M.); happy.msiza@yahoo.com (N.H.M.)

2 Food Security and Safety Niche Area, Faculty of Natural and Agricultural Sciences, North-West University, Mmabatho 2735, South Africa

3 Department of Animal Production, Fort Cox Agriculture and Forestry Training Institute, Middledrift 5685, South Africa; bethwellm@gmail.com

* Correspondence: ravhuhalike@gmail.com (K.E.R.); onkehawu97@gmail.com (O.H.)

**Abstract:** Globally, there have been differing views on whether the proliferation of invasive species will be of benefit as a livestock feed source or it will have detrimental effects on the ecosystem. The infestation of invasive plants such as *Prosopis* species does not only affect the groundwater levels but also threatens the grazing capacity and species richness of most of the semi-arid areas around South Africa. Though *Prosopis* is invasive, it is however of good nutritive value and can serve as an alternative source of protein and minerals for livestock during the dry season. Bush encroachment by browsable invasive species can be controlled through biological methods by using organisms such as livestock. The utilisation of *Prosopis* through browse benefits livestock production and at the same time reduces its spread, thereby preventing possible environmental harm that may arise. Although several studies have been carried out globally on the assessment of the *Prosopis* species' nutritive value and also on the threat of this invasive species to the environment, there is a need to update the state of knowledge on this species, particularly in the context of the semi-arid areas of South Africa where the dry season is characterised by less herbage of poor quality. It is therefore critical to understand whether *Prosopis* is a beneficial invader, or a detriment that needs to be eradicated. This review will contribute knowledge towards finding practical solutions to controlling *Prosopis* species and whether utilising *Prosopis* as a feed source will limit its spread and result in a vegetation structure where *Prosopis* becomes part of the ecosystem with limited detrimental impact. This means that the several components of the species such as nutritive value and the negative impact associated with this plant species along with the means to control its spreading must be well understood to recognise the plant species' vital contribution to the ecosystem.

**Keywords:** *Prosopis* species; livestock; nutritive value; invasive species; environment; semi-arid

## 1. Introduction

In South Africa over the past half-century, various species of deciduous, leguminous thorn tree species have been introduced for various purposes, such as timber, firewood, bark for tannins, medicines, windbreaks, edible products for humans, and fodder for animals [1,2]. These trees include *Acacia mearnsii*, *Opuntia ficus-indica*, and *Prosopis* species. The genus *Prosopis* has several species and hybrids, and in South Africa, the dominant ones are *Prosopis glandulosa* (Honey mesquite) and *Prosopis velutina* (Velvet mesquite) [3]. The *Prosopis* species were initially introduced to South Africa from South, Central, and North America in the late 1880s, mainly to provide fodder (pods in drought years), shade for livestock, windbreak, wood for fuel, timber for furniture, and a nectar source for honey production [4–7]. The plant was of great value to all stakeholders until the 1960s [8,9], before its

negative invasive impact on ecosystem services, biodiversity, and local people's livelihoods was observed [10,11]. The species has invaded arid and semi-arid parts of Southern Africa, as well as other parts of the world [12]. The high invasive capacity is derived from its vigorous growth and high seed production and efficient dispersal mechanism, while the absence of natural seed-eating insects preserves the seed for extended periods [13]. In South Africa, Zachariades et al. [7] estimate that 1.8 million ha is covered by *Prosopis*. Various *Prosopis* species were introduced for various purposes such as shade provision for livestock and windbreak, leaves, and pods for feeding livestock. The deliberate planting led to numerous sources of seeds, which were spread distant and wide, both endozoochorously and through flooding occasions. *Prosopis* is an invasive tree species known for invading several millions of hectares of land in the Western Cape, Northern Cape, Free State, and North-West Provinces of South Africa, creating extensive, invulnerable thickets over vast regions [14]. Other than overwhelming the grazing land, devouring excessive amounts of groundwater, and reducing biodiversity, *Prosopis* is a very noxious invader, with areas of high infestation resulting in surrounding indigenous plants failing to deliver valuable ecosystem services for that ecological niche [15].

Besides its invasive problem, *Prosopis* still provides some nutritional benefits to livestock [16], and therefore, any control programme should not ignore its contribution to the smallholder livestock farmers in semi-arid areas. Furthermore, beneficial ecosystem services such as the reduction of soil erosion are obtained from this invasive plant [17]. It is therefore important to review the current knowledge on both the invasive and positive impact of *Prosopis* in semi-arid areas of Southern Africa. The information will aid in developing sustainable management strategies for the benefit of both biodiversity conservationists and livestock producers.

## 2. The Expansion of Invasive Species

Numerous tree plants have amplified their ranges within the previous few centuries as a result of human activities. Dunbar and Facelli [18] stated that several invasive species introduced into an area are considered pests for agricultural industries, as they pose an economic risk to these industries. Various writers have considered the expanding number of invasive species as a main, vital part of global change, because of their great ability to modify the essential efficiency, hydrology, nutrient cycling (soil improvement), and decomposition in the ecosystem [19–21]. Shackleton et al. [20] indicated that while appreciating the existence of the species, in order to manipulate the invasion of this species, there is a need to introduce the capital (namely natural, social, human, physical, and financial capitals) in order to reduce human vulnerability to natural disasters. Vitsousek et al. [19] reported that numerous invasive species biodiversity associations have consolidated the invasive plant tree species in their primary activities and have defined rules for their monitoring and annihilation. This includes partnering with relevant government entities (for policy and legislation) and other institutions, together with the land users [20,21].

Shackleton et al. [20] highlighted that even though some of these invasive species can be beneficial, there are some detrimental aspects that can create vulnerability in social–ecological systems. The detrimental and beneficial aspects of *Prosopis* invasive plant species on a widespread extent, especially on livestock and underground water, have been reported in numerous locations of the world [22–24]. Hence, the invasive woody alien plant species, as non-native organisms that increase from the point of introduction and become much more abundant, have a great potential to cause harm to the environment, as they are the key drivers of environmental change, disrupting ecosystem functioning, being detrimental to grazing lands, and tending to threaten the native biological diversity (being the main causes of biodiversity losses around the world), economics, and human and animal health [11,25–28].

## 3. Different *Prosopis* Species

In South Africa, three different *Prosopis* plant species were introduced from North, South, and Central America in the last 1800s, namely, *P. glandulosa*, *P. chilensi*, and *P. velutina* (Figure 1) [29]. The study of Visser [3] reported that of the above-mentioned species, as well as their crossbreeds, there are only two species that prosper in South African environmental conditions, specifically *Prosopis velutina* and *Prosopis glandulosa* var. *torreyana*. Wild and du Plessis [8] stated that due to the nature of the species and its uncommon ability to adjust to extraordinary climate conditions, together with the potential of the high protein content of its pods, it can be used as a protein supplement for livestock during the dry season. Some of these *Prosopis* plant species are confirmed class II intruders, meaning that they are permitted to be grown in differentiated regions by allowing (permit) holders for prudent utilisation such as charcoal, building resource materials, and erosion control, as well as for medicinal purposes [11,24,30].

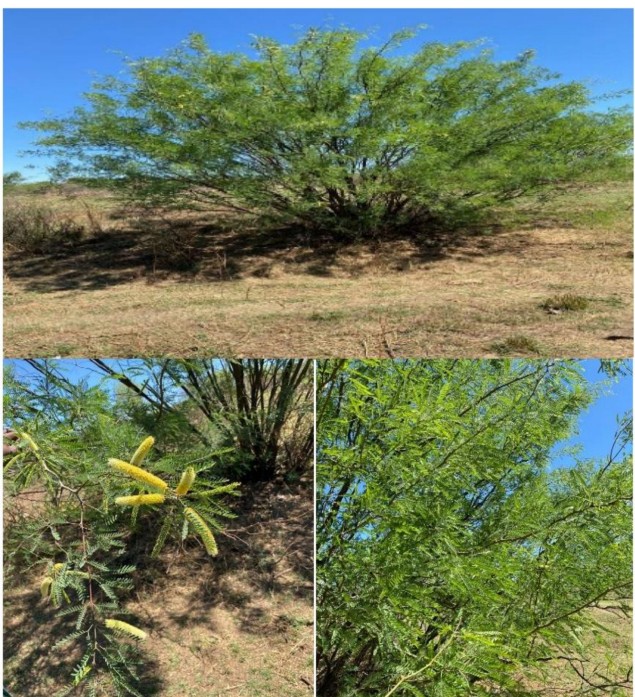

**Figure 1.** *Prosopis velutina*, Mafikeng municipality NW province. Photo taken by K.E. and H.S.

These species and their crossbreeds (hybrids) have been recorded as invasive species in terms of the National Environmental Management, Alien and Invasive Species Regulations (NEMBA) act (Act No. 10 of 2004). Early hybridisation between *P. glandulosa* var. *torreyana* and *P. velutina* has created a very intrusive hybrid due to the effect of "hybrid vigour". Specifically, the species were precisely recorded as Category 1b species within the Western and Eastern Cape, Free State, and North-West Provinces of South Africa, which implies that they need to be controlled or overcome and removed in any conceivable way [31]. Hence, in the Northern Cape Province, they are categorised as Status or Category 3, which implies that they may stay where they are presently, excluding riparian regions, where they will be considered as Category 1b species [32]. No trade, proliferation, or planting is permitted in any of the listed provinces. The rules and regulations do not apply in any of the provinces of South Africa that are not listed.

## 4. The Habitat of *Prosopis* Species around South Africa

The genus has invaded several hectares of the western half of South Africa, forming extensive and impenetrable thickets over vast areas [14,33]. The study of Nel et al. [34] reported that numerous plant species have amplified their ranges within the past few

centuries due to human activities. The increase in *Prosopis* species and the formation of extensive infestations have been widely escalated by wild and domestic animals, which mostly feed on ripe seed pods and scatter scarified seeds [12,28]. The *Prosopis* species distribution in South Africa is either by pods carried by flooding events and softened by water or dispersed by animals under the form of scarified seeds [24].

It is estimated that the spreading rate of *Prosopis* species in South Africa ranges from 18 to 40% per annum [24]. Van den Berg [35] stated that once the *Prosopis* species have set up in rangeland, the density of the infestations rapidly spreads at yearly rates of 3–10%. In the Northern Cape, Van den Berg [35] showed that the average annual rate of spread of *Prosopis* is very high, being approximately 15% in upland areas and up to 30% in riparian areas. Several estimations of provinces invaded with *Prosopis* species have been made over the decades [14,33]. With the use of biome-based procedures when ranking invasive plant species in South Africa [36], Robertson [37] stated that in South Africa, *Prosopis* plant species were ranked the second species in the Nama-Karoo biome and the third in the Succulent Karoo biome. Martin [38] reported that *Prosopis* seed may last a long time and may gain in mass over time to sizeable seed storage, which can endure for a minimum of 20 years without deteriorating. As per the study of Roberts [39], the measure of seed storage in South Africa changes over the distributional range of *Prosopis* species and is influenced by the existence or non-existence of animals, with accumulations of as numerous as 2500 seeds/m$^2$ in some few regions.

## 5. Ecophysiology, Drought, and Salt Tolerance

Several authors highlighted the adaptability of this species in dry areas with more saline soils unsuitable for cultivation [12,40,41]. For example, Lauenstein et al. [42] stated that *Prosopis* species, especially *flexuosa*, grow and develop in a broader range in the flatlands where there is no additional water contribution. This could be linked to moderate plasticity at physiological and xylem anatomical levels and a positive absolute value in key drought tolerance characteristics. According to Villagra et al. [43], some of these species can survive the desert steppe with extremely severe low temperatures during winter times. *Prosopis* species have a defence mechanism or a system against drought strain, which involves alterations in gas exchange, stomata opening, osmotic adjustment, and leaf area [44]. *Prosopis* species are extremely tolerating salt [45], e.g., *P. juliflora*, *P. tamarugo*, *P. laevigata*, *P. alba*, and *P. pallida*, and can grow in saline soil regimes comparable to seawater [46–48].

## 6. The Negative Impact Associated with *Prosopis* Species

The negative effects of *Prosopis* invasions to the environment and biodiversity include the reduction in plant species richness, density, and diversity in arid areas [49], as well as increased local tree mortality due to increased competition for water, nutrients, and land with existing local vegetation [22,26,50–52]. The ecosystem activities, i.e., water supply, soil quality, and grazing areas, have been negatively impacted by *Prosopis* invasions, resulting in a range of negative results for native farmers [24,53–55]. Even though it can provide supplement protein in the dry winter season, Wise et al. [24] reported that the utilisation of *Prosopis* species is very limited or suppressed due to the existence of anti-nutritional factors, which tend to be poisonous to the livestock if consumed in large quantities.

## 7. *Prosopis* as an Invasive Species

Richardson et al. [56] defined an invasive species as naturalised plants that produce reproductive offspring, often in very large numbers, at considerable distances from parent plants (approximate scales: >100 m; <50 years for taxa spreading by seeds and other propagules; >6 m/3 years for taxa spreading by roots, rhizomes stolons, or creeping stems), and thus have the potential to spread over the considerable area. Currently, the invasive species are of concern for biological conservationists, environmentalists, and ecologists around the world [57]. According to the study of Shackleton [11], these biological invasions

create a vital hazard to biodiversity and have been acknowledged as a major non-climatic driver of global change. *Prosopis* (mesquites) species were mostly introduced in arid and semi-arid parts of South Africa in the late 1800s by the Department of Agriculture and Rural Development Corporation with the main motives that they will benefit either livestock or humans together with the ecosystem [6].

According to Pasiecznik [12], several key factors that are favourable for invasive species to dominate over the area include climate change, land-use changes, and competitive ecological advantages. For instance, in South Africa, the widespread occurrence of *Prosopis* invasive plant species takes place mostly in the areas where there is a scant herbaceous layer available and where the conditions for establishment and germination are favourable [6,58]. According to Harding [59], seed production is predicted at 600,000 to 1,000,000 seeds per mature tree each year. It was highlighted that those seeds are most likely to sprout when they are scoured, as they pass throughout the digestive tract and are released into the humid faeces of ruminants [60].

Invasive alien *Prosopis* (mesquite) is known to suppress the germination and establishment of indigenous vegetation and forms a dense population. According to Lowe et al. [61], several initiatives to treat, control, and manage the invasive plant species have been practised in communities.

## 8. *Prosopis* Ecosystem Services

The *Prosopis* plant species have long been considered a significant plant species in arid and semi-arid regions [62]. These plant species carry out a multipurpose role that includes soil conditioning (improving soil fertility by balancing N, K, and P concentrations in the soil), controlling soil erosion, providing fuel energy resources, and providing wood for furniture and timber for construction [30,63]. With regard to indigenous knowledge on species ecosystem services, Shackleton and Shackleton [64] found that invasive species such as *Prosopis* species produce good high-value charcoal whose heartwood is very well built, durable, and high-quality biofuel energy that tends to burn very well. The plant species also provide fencing for dwelling compounds and farmlands, some shelter for both animals and human beings, fodder for livestock from its fruits and leaves, windbreak for protecting against heavy winds, and shade from sunburn [12,24,64].

### 8.1. Prosopis as a Feed Source for Livestock Production

The shortage of sufficient and high-grade forage is a key restriction in tropical livestock production [65]. As indicated, in several parts of the tropics, the utilisation of browse species as feed for ruminants is growing, especially when the amount and value of pastures are poor for a long time [66–69]. *Prosopis* pods play a great beneficiary role in livestock production, society, and the general economies in arid areas [70]. Several uses have been reported over the years for *Prosopis* plant species such as animal feed [71–73], due to their high carbohydrate and protein content and their bioactivities, along with their medicinal properties [74–79]. Due to their higher nutritive value than pasture, the pods and leaves of *Prosopis* species are very edible and are consumed voluntarily by goats, sheep, camels, and cattle. Pods can also be fed to monogastric animals as well [12]. Baptista [80] stated that the leaves of *P. glandulosa* had relatively low concentrations of fibre: 32–43% for neutral detergent fibre (NDF) and 23–33% for acid detergent fibre (ADF). *P. juliflora* leaves are rich in crude protein (CP) (roughly 20%) and low in fibre (23.4%) levels, and they are usually not palatable because of flavonoids, polyphenols, and tannins [81,82]. Pods of *Prosopis* containing CP (12%), EE (2.6%) CF (25.4%), and ash (0.4%) were reported by Mahgoub et al. [71], and Sawal et al. [62] found that *Prosopis* pods are highly palatable and have high nutritive value, comprising protein (7–22%), crude fibre (11–35%), fat (1–6%), ash (3–6%) and carbohydrates (30–75%) [83]. Phosphorus, Mg, and Ca are the most critically required minerals by animals, as reported by Kebede [84]. Al-Harthi [85] detailed that the pods contained a mineral concentration of 0.66% Ca, 0.20% total P, 764 ppm Fe, 69.4 ppm Zn, 33.9 ppm Mn, 36.1 ppm Cu, 21.7 ppm Cr, 7.4 ppm Cd, 9.8 ppm Ni, and 28.2 ppm Pb.

*Prosopis* pods have a satisfactory amount of minerals [85] that could be of good productivity and display no signs of insufficiencies.

Eldaw [86] stated that the proteins of *Prosopis* pods carry almost all the essential amino acids in higher concentration than found in leaves. According to Astudillo et al. [87], the amino acid concentration of leaves from six mesquite species had similar values to those of lucerne. Mohamed et al. [88] also stated that the rich content of *Prosopis* pods with a concentration of energy, minerals, and proteins gives strong evidence that most *Prosopis* species can be the possible feed tree that can meet the animal's nutrient requirements for the sustainability of animal production. Table 1 shows the chemical composition of *Prosopis* species leaves and pods (% of DM), and Table 2 shows the nutritive value of *Prosopis velutina* leaves harvested around North West province and analysed in North West University Animal Science Laboratory. Table 3 shows the mineral concentration of *Prosopis* species pods, and Table 4 shows amino acids of *Prosopis* species leaves, pods, and seeds.

### 8.2. Prosopis Species for Medicinal Purposes

Various reports are highlighting that *Prosopis* species have provided treatments for several years, treating various ailments [89–91]. Over many years, the plant species have been utilised for traditional medicines, usually using their pods, leaves, roots, and seeds for treating various maladies [92,93]. In South Africa, *Prosopis* pods have been utilised to stabilise blood sugar levels in humans as an indigenous medicine (manna) [94]. Bioactive compounds found in *Prosopis* species play a substantial role in indigenous medical systems [95,96]. Technically, the plant species has a vast history in medical practices and its versatile uses in local areas [97,98].

**Table 1.** Chemical composition of *Prosopis* species (% of DM).

| Species | PP | DM | CP | CF | Ash | OM | NDF | ADF | ADL | References |
|---|---|---|---|---|---|---|---|---|---|---|
| *P. chilensis* | P | 82.0 | 7.1 | 12.6 | 3.0 | - | - | - | - | [99] |
| *P. chilensis* | L | - | 18.3 | 25.1 | 4.5 | - | 37.5 | 28.8 | - | [100] |
| *P. juliflora* | P | 88.4 | 18.5 | - | 5.2 | 83.2 | 51.8 | 29.8 | 3.2 | [85] |
| *P. juliflora* | L | 92.5 | 10.4 | 23.7 | 9.1 | - | 48.4 | 35.1 | 13.1 | [82] |
| *P laevigata* | P | 92.5 | 39.4 | 7.6 | 5.1 | - | 32.9 | 11.8 | - | [101] |
| *P. velutina* | L | - | 20.2 | 27.0 | 5.5 | - | 41.8 | 33.1 | - | [100] |
| *P. pallida* | P | 85.9 | 9.1 | - | 3.9 | - | - | - | - | [12] |
| *P. cineraria* | P | 91.0 | 13.5 | 14.3 | 5.2 | - | - | 21.4 | - | [102] |
| *P. cineraria* | L | 93.2 | 10.7 | - | 13.8 | 86.2 | 45.8 | 29.0 | - | [103] |

PP: plant part, P: pods, L: leaves, DM: dry matter, CP: crude protein, CF: crude fibre, OM: organic matter, NDF: neutral detergent fibre, ADF: acid detergent fibre, ADL: acid detergent lignin.

**Table 2.** Chemical composition (g/kg DM, unless otherwise stated) of *Prosopis velutina* leaves harvested around Mafikeng North West province, South Africa.

| DM g/kg | OM | CP | NDF | ADF | ADL | References |
|---|---|---|---|---|---|---|
| 909.7 | 884.2 | 364.4 | 255.2 | 251.2 | 157.4 | [104] |
| 967.89 | 925.01 | 117.9 | 515.2 | 345.1 | 232.7 | [105] |

DM: dry matter, OM: organic matter, CP: crude protein, NDF: neutral detergent fibre, ADF: acid detergent fibre, ADL: acid detergent lignin.

**Table 3.** Mineral concentration (ppm DM) of *Prosopis* species Pods.

| Species | Ca | P | K | Mg | Cu | Fe | Na | References |
|---|---|---|---|---|---|---|---|---|
| *P. chilensis* | 8000 | 1900 | 18500 | 1800 | 12 | 55 | 996 | [88] |
| *P. juliflora* | 5000 | 2000 | 9000 | 760 | 40 | 99 | 51 | [85] |
| *P. glandulosa* | 60 | 2280 | 540 | 40 | | | | [16] |
| *P. pallida* | 800 | | 26500 | 900 | | 300 | 1100 | [83] |

Ca: calcium, P: phosphorus, K: potassium, Mg: magnesium, Cu: copper, Fe: iron, Na: sodium.

**Table 4.** Amino acid levels of *Prosopis* species on different plant parts.

| Species | PP | Thr | Val | Met | Ile | Leu | Phe | His | Lys | Arg | Try | Pro | Asp | Glu | Units | Ref |
|---|---|---|---|---|---|---|---|---|---|---|---|---|---|---|---|---|
| *P. pallida* | P | 4.68 | 7.80 | 0.57 | 3.26 | 7.94 | 2.98 | 1.99 | 4.26 | 4.82 | 0.89 | 23.40 | 8.51 | 10.07 | g/100 g | [12] |
| *P. africana* | S | 2.25 | 4.13 | 1.86 | 3.46 | 13.26 | 4.82 | 32.16 | 2.77 | 3.62 | 3.24 | 4.22 | 4.58 | 4.68 | mg/100 g | [106] |
| *P. alba* | L | 1.20 | 1.26 | 0.30 | 1.20 | 1.58 | 0.34 | 0.80 | 1.68 | 5.50 | | 4.30 | 3.17 | 3.48 | g/16 gN | [87] |
| *P. chilensis* | P | 8.81 | 13.76 | 4.14 | 40.41 | 19.07 | 12.54 | 9.66 | 14.75 | | | | 21.48 | | g/100 g | [107] |
| *P. laevigata* | S | 29.8 | 34.8 | 9.1 | 29.2 | 69.1 | 35.6 | 24.2 | 54.8 | 112.2 | | 62.6 | 83.4 | 1172 | mg/g | [108] |
| *P. chilensis* | L | 2.81 | 7.11 | 1.31 | 4.51 | 8.25 | 3.20 | 4.44 | 2.94 | | | | 8.88 | | g/100 g | [107] |
| *P. julifora* | P | 0.46 | 0.71 | 0.20 | 0.44 | 1.33 | 0.71 | 0.55 | 0.81 | 2.69 | 0.22 | | | | g/100 g | [109] |

PP: plant part, S: seeds, L: leaves, P: pods, Thr: threonine, Val: valine, Met: methionine, Ile: isoleucine, Leu: leucine, Phe: phenylalanine, His: histidine, Lys: lysine, Arg: arginine, Try: tryptophan, Pro: proline, Asp: aspartic acid, Glu: glutamic acid, Ref: references.

## 9. The Anti-Nutritional Factors Associated with *Prosopis* Species

Ehsen [110] stated that anti-nutritional factors (ANFs) are secondary plant metabolites and are considered to be biologically active substances. The fruits, seed, and other plant parts produce these substances [111,112]. A study conducted by Anhwange et al. [106] revealed that *Prosopis* species contain ANFs, i.e., saponins, alkaloids, tannins, and oxalates, in varying quantities. The utility of *Prosopis* species is limited as animal feed by the existence of ANFs. According to Aganga and Tswenyane [112], ANFs reduce livestock productivity, but they can cause toxicity or confinement if animals eat large amounts of feed rich in these substances.

Saponins are glycosides comprising a polycyclic aglycone of any C27 steroid or C30 triterpenoid bound to carbohydrate [112,113]. They occur in *Prosopis* species and other various distinct plants. According to Thomas [113], saponins have a characteristic unpleasant taste, foam in water, and may induce red blood cell haemolysis. The levels of ANFs in *Prosopis* pods were reported for saponin (317 mg/100 g), total phenol (640 mg/100 g), tannin (860 mg/100 g), and phytic acid (181 mg/100 g) [86,114]. The trypsin inhibitors such as haemeaglutins are heat liable and are concentrated on the seed for *P. glandulosa* as, as reported by Eldaw [86]. According to Thomas [113], tannins are complex polyphenolic plant compounds soluble in polar solutions and capable of precipitating several biomolecules, including carbohydrates, minerals, and proteins. In high proportion, tannins in *Prosopis* leaves have detrimental effects on the digestibility of CP and DM, and they lower the retention of nitrogen. The leaves of *Prosopis* contain 2.2% tannins per DM, and young leaves have a greater level of tannins than older leaves [115].

According to Panche [116], flavonoids belong to a class of secondary plant metabolites with a polyphenolic structure commonly found in plants and are a vital class of natural products. Several studies have confirmed the presence of flavonoids in the *Prosopis* species. Young et al. [117] found luteolin, myricetin, and quercetin in the pods of *Prosopis alba*. Amorowicz [118] described legume seeds as a very good flavonoid source for apigenin, quercetin, daidzein, kaempferol, and genistein. Diaz-Batalla [108] stated that the seed of *Prosopis* is a good source of apigenin and a vital active constituent with positive health effects on animals. Table 5 demonstrates the levels of ANFs of different *Prosopis* species.

**Table 5.** Anti-nutritional factors (% DM) of *Prosopis* species plant parts.

| Species | PP | Tannins | Saponins | Oxalates | Flavanoids | Alkaloids | Nitrates | Phenols | Source |
|---|---|---|---|---|---|---|---|---|---|
| *P. glandulosa* | L | 0.646 | 1.693 | 0.721 | 0.755 | | 0.356 | 0.127 | [110] |
| *P. julifora* | P | 0.973 | 0.393 | | | 0.08 | | 0.582 | [119,120] |
| *P. cineraria* | L | 5.751 | 1.324 | 0.361 | 1.113 | | 0.224 | 0.331 | [110] |

PP: plant part, L: leaves, P: pods

## 10. Livestock as a Tool to Control Invasive Species

Many studies have been conducted in an attempt to reduce and control the increase in invasive species. Livestock grazing in low invasive species abundance and separate

species zonation common in wetland ecosystems may permit the superior achievement and targeted control of invasive species [121]. Zedler and Kercher [122] suggested that livestock could be useful resources for handling the influences and increase in invasive species in marshes where monoculture-forming invasive species are ordinary and drive large-scale ecosystem alteration. On the other hand, livestock is considered as one of the chief contributors to the spread of invasive species, as they can introduce pods from outside the area [53].

In the African continent and other continents, usage of livestock to manage invasive species has been fundamentally limited to world grasslands, where this technique has been met with diverse success [121,123]. Although small stock alone as a treatment cannot successfully eradicate invasive species, few authors have documented the use of goats. Mayo [124] used goats to control *Sericea lespedeza*, and a reduction in seed production was witnessed. Results by Rathfon et al. [125] suggested that goats give an effective and environmentally friendly method to control invasive species. According to Nyamukanza and Scogings [126], constant browsing by goats of *Acacia karroo* sprouts when young will decrease the number of regrowth coppices, halting the species' expansion in semi-arid regions of South Africa. According to Esselink et al. [127], livestock strongly limit invasive species spread and enable the development of shorter grasses in its natural environment. The study of Reiner and Craig [123] addressed the statement that livestock grazing is a conservation-compatible land use on spreads with conservation easements. In the recent past, many studies have been carried out with the aim of controlling invasive species, but with poor results. Surprisingly, there is a paucity of information in the literature concerning the control of invasive alien species. Livestock grazing is still considered the main solution to this problem; however, there is little research on the control of browsable alien invasive plants.

## 11. Summary

The control of invasive alien species is based on their contribution to the ecosystem and also on the negative impact associated with the species. For the development of better control, approaches, well-trained personnel, and knowledge of the species and the spreading process are very important. As far as *Prosopis* is concerned, their nutritive value to livestock makes it a valuable component of the rangelands for resource-constrained communal farmers. It is therefore important to develop utilisation strategies that consider the effective age or stage of development for the maximum control of spread and are also of benefit to ruminants. Therefore, managing the spread of these invasive species can be accomplished by the use of livestock as biological control while improving the productivity of ruminant animals. There is also a need to balance its use as a protein supplement and its negative impact on herbaceous biomass production. Additional control strategies such as physical ones can be applied to reduce the number of *Prosopis* plants to a level where optimum herbaceous biomass for livestock production can be achieved and the potential impact on soil erosion is minimised. Hence, these invasive *Prosopis* species control will assist in maximising the grazing capacity while maintaining the species diversity in arid and semi-arid environments.

**Author Contributions:** K.E.R., H.S.M., B.M., O.H., N.H.M. contributed equally to the writing of the review article. All authors have read and agreed to the published version of the manuscript.

**Funding:** This research received no external funding.

**Institutional Review Board Statement:** Not applicable.

**Informed Consent Statement:** Not applicable.

**Data Availability Statement:** Not applicable.

**Conflicts of Interest:** The authors declare no conflict of interest.

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
