# Peer review of "Prosopis Species—An Invasive Species and a Potential Source of Browse for Livestock in Semi-Arid Areas of South Africa"

_sustainability, doi:10.3390/su13137369_

Round 1
Reviewer 1 Report
General
Balancing potential ecosystem services of invasive species and their negative effects is an actual hot topic. Many authors dispute about these issues and there is still no unanimous view whether this species should be treated always as aliens or could be also a source of several positive contributions for the ecosystem or humans.
The paper examines the both aspects for the presence of Prosopis species in South Africa.
Along the paper there are repetitions of some known general knowledge about alien invasive plants. It can be cleared and references unified.
Specific comments
Introduction:
Gives a clear perspective of Prosopis invasive effects. What is missing is the legal aspect of its status. Some information is given in the Chapter 4. Different Prosopis species. But it has to be summarized here too. Is Prosopis on the invasive list in South Africa? Please add these information.
Chapter 6.
Lines: 160-162 The definition is not correct. It is related to alien, but not invasive. Please modify, even if taken from literature.
Lines: 184-185 This general statement has been repeated several times. It can be deleted and reference added in some previous sentence with the same meaning.
The chapter 7. Is entitled “Importance of Prosopis species in the ecosystem”. Still in the chapter is elaborated in which ecosystem services is Prosopis useful. For example the use as charcoal is not of use for the ecosystem in general. Maybe the more appropriate title of the chapter could be “Prosopis ecosystem services” or similar. These different ecosystem services can be grouped according to Millennium Ecosystem Assessment.
Summary
In order to manage this species its spreading must be well understood to recognise the plant species’ vital contribution to the ecosystem. The paper collect and systematizes knowledge on Prosopis species. It is interesting to general readers and useful to professionals.
Author Response
We would like to thank the Editors and reviewers for taking time to go through our review article and providing constructive comments. We have gone through all the comments and our carefully considered responses are appended below. We hope that our responses are satisfactory, however we stand ready to make further changes should these be required.
Reviewer 1 |
|
Comment |
Response |
Balancing potential ecosystem services of invasive species and their negative effects is an actual hot topic. Many authors dispute about these issues and there is still no unanimous view whether this species should be treated always as aliens or could be also a source of several positive contributions for the ecosystem or humans. |
Thank you for your comments. We have incorporated the invasive species into the browse species. |
Along the paper there are repetitions of some known general knowledge about alien invasive plants. It can be cleared and references unified. |
Thank you for noting that we have tried to clear all repetitions by deleting some of the sentences on the manuscript as seen on track changes. However, we are willing to make any chances should the reviewer feel that we need to make changes. |
Introduction: Gives a clear perspective of Prosopis invasive effects. What is missing is the legal aspect of its status. Some information is given in the Chapter 4. Different Prosopis species. But it has to be summarized here too. Is Prosopis on the invasive list in South Africa? Please add this information. |
Thank you for your comment. We have now included information on the legality of Prosopis there by attaching the National Environmental Management, Alien and Invasive Species Regulations (NEMBA) act (Act No.10 2004). As stated on Chapter 4 we have indicated that these species are being considered as an invasive species as category 1b in other provinces and category 3 of some provinces in South Africa of species in line with the NEMBA act 2004. |
Chapter 6. Lines: 160-162 The definition is not correct. It is related to alien, but not invasive. Please modify, even if taken from literature. |
Thank you and noted, in Lines 160-162 the correct definition has been provided. The correct definition is stated as follows: An invasive species is an indigenous or exotic plant species existing out of its natural distributional rangelands or semi natural ecosystems and gives rise to the particular management problems especially when it emerges where it is not wanted and threatens the natural biological diversity |
Lines: 184-185 This general statement has been repeated several times. It can be deleted and reference added in some previous sentence with the same meaning. |
Thank you for the comment. Lines 184-185 have been adjusted and have been deleted and references were added. |
The chapter 7. Is entitled “Importance of Prosopis species in the ecosystem”. Still in the chapter is elaborated in which ecosystem services is Prosopis useful. For example, the use as charcoal is not of use for the ecosystem in general. Maybe the more appropriate title of the chapter could be “Prosopis ecosystem services” or similar. These different ecosystem services can be grouped according to Millennium Ecosystem Assessment. |
Thank you for the advice. We have renamed the title Prosopis ecosystem services |
Summary: In order to manage this species its spreading must be well understood to recognise the plant species’ vital contribution to the ecosystem. The paper collects and systematizes knowledge on Prosopis species. It is interesting to general readers and useful to professionals. |
Thank you very much for the positive appraisal of our work. |
Reviewer 2 Report
Dear Authors,
Although interesting I found the paper difficult to read and somewhat confusing. Therefore, I’d suggest reorganizing the paragraphs in which the work is structured avoiding the numerous repetitions that are present in the text. In my opinion, the sequence of presentation of the paragraphs themselves should also be carefully reconsidered. In any case, it is necessary to report what the authors believe is useful and possible to do in relation to the problem of limiting the spread of Prosopis species, reporting in the conclusions, currently not present in the text, clear operational indications on the choices to be made and on the most suitable strategies to follow. It would also be useful to estimate the diffusion of the problem on a territorial scale. An initial botanical description of the mainly widespread species, as well as their ecophysiology, geographic distribution, habit, drought and salt tolerance, etc. should be included. It would be useful for those readers not particularly familiar with the subject. The most relevant research needs should be also reported.
Finally, it is advisable to carefully review the most updated bibliography on the subject such as:
- The role of invasive alien species in shaping local livelihoods and human well-being: A review, by Ross T. Shackleton, Charlie M. Shackleton and Christian A. Kull, in: Journal of Environmental Management 229 (2019) 145–157.
- Local knowledge regarding ecosystem services and disservices from invasive alien plants in the arid Kalahari, South Africa, by Sheona E. Shackleton and Ross T. Shackleton, in: Journal of Arid Environments 159 (2018) 22-33.
- Towards a national strategy to optimise the management of a widespread invasive tree (Prosopis species; mesquite) in South Africa, by Ross T. Shackleton, David C. Le Maitre, Brian W. van Wilgen and David M. Richardson, in: Ecosystem Services 27 (2017) 242–252.
- The economics of landscape restoration: Benefits of controlling bush encroachment and invasive plant species in South Africa and Namibia, by William Stafford, Catherine Birch, Hannes Etter, Ryan Blanchard, Shepherd Mudavanhu et Al., in: Ecosystem Services 27 (2017) 193–202.
Author Response
We would like to thank the Editors and reviewers for taking time to go through our review article and providing constructive comments. We have gone through all the comments and our carefully considered responses are appended below. We hope that our responses are satisfactory, however we stand ready to make further changes should these be required.
Reviewer 3 |
|
Although interesting I found the paper difficult to read and somewhat confusing. Therefore, I’d suggest reorganizing the paragraphs in which the work is structured avoiding the numerous repetitions that are present in the text.
|
Thank you for the suggestion. We have reorganized all the paragraphs to be within the scope of each heading whereby in other headings we created subheadings as seen in track changes to avoid confusion and it will improve paper readership. |
In my opinion, the sequence of presentation of the paragraphs themselves should also be carefully reconsidered.
|
Thank you for the comment. The sequence of the presentation of paragraphs have been reconsidered carefully as per track changes. |
In any case, it is necessary to report what the authors believe is useful and possible to do in relation to the problem of limiting the spread of Prosopis species, reporting in the conclusions, currently not present in the text, clear operational indications on the choices to be made and on the most suitable strategies to follow. It would also be useful to estimate the diffusion of the problem on a territorial scale. |
We have revised both chapter 9 as a tool to control invasive species and chapter 11 summary. This revision was done in order to address the useful solution in relation to the problem of spreading of the Prosopis species. All these useful solutions were stated in the conclusion/summary. |
An initial botanical description of the mainly widespread species, as well as their ecophysiology, geographic distribution, habit, drought and salt tolerance, etc. should be included. |
Thank you, all these subheadings are now included in the review as seen on track changes and subheadings that were already there. We are still willing to make any changes should the reviewer not feel satisfied with our amendments. |
It would be useful for those readers not particularly familiar with the subject. The most relevant research needs should be also reported. |
Thank you for the suggestion |
We have gone through the copy in order to modify the whole copy, and changes were made from the title up to conclusion as seen on track changes. We are still willing to make any changes should the reviewer not feel satisfied with our amendments. |
Round 2
Reviewer 2 Report
Dear Authors,
I appreciated your efforts to make the paper easier to read. However, in my opinion, there is still a lot to do in terms of reorganizing the paper and corrections to be made. There are, for instance, numerous repetitions that should be avoided and too much bla-bla that must be eliminated: for example #8.1 and 8.2 could be profitably merged and synthetized. Furthermore, there are still some research papers that I’ve already suggested are worthwhile to be included in the bibliography and adequately commented by you.
Additionally, please, take note of the following suggestions:
Line/s:
- 49: change “universe” with “world”
- 60: Prosopis
- 70: this should be the right place for introducing and summarizing the most relevant and recent review literature (e.g.: The role of invasive alien species in shaping local livelihoods and human well-being: A review, by Ross T. Shackleton, Charlie M. Shackleton and Christian A. Kull, in: Journal of Environmental Management 229 (2019) 145–157).
- 78: what do you mean by: numerous invasive species association on biodiversity associations have consolidated….?? Please check and rephrase it.
- 82: Did you mean “extent”?
- 90: Don’t you think it would be better to first describe Prosopis spp.? (i.e. move #4 to #3)
- 91: In any case: “The species has invaded several…” change to “The genus has invaded several…”
- 96-112: The Prosopis species distribution in South Africa is facilitated either by pods that were carried through by flooding events where they would soften in the water and release seeds for germination and softened by water or dispersed by the time when animals consume the ripe seed-pods which they then attempt to digest and excrete the scarified seeds over a dispersed area under the form of scarified seeds [24]. The excreted scarified seeds would then germinate in their new environments.
- 114-115 & 116-117: avoid repetition.
- 132: “crossbreeds; there”: delete semicolon
- 188-191: for a clear-cut definition of “invasive”, please, refer to: RICHARDSON D.M., PYŠEK P., REJMÁNEK M, BARBOUR M.G., PANETTA F.D. & WEST C.J., 2000 – Naturalization and invasion of alien plants: concepts and definitions. – Diversity Distrib., 6: 93-107.
- 191-192: what do you mean by: “…are a vital for biological conservationists,” ?
- 201-204: this seems to me an unnecessary repetition
- 208: “in theareas” insert space
- 210-212: again, unnecessary repetition
- 221-223: Move to the conclusions
- 225-228: Move to #5
- 238-331: Merge and synthetize in only one chapter
- 248: (citation): use the form 71-76
- 296-299: too much bla-bla….
- 333-334: Unclear sentence: please, rephrase it
- 334: as for 248: (94-96)
- 432: Velutina not Vetulina!
- 432: avoid repeating g/Kg…
- Table 1 & 3 Pallida not Palida!
- Table 2: tables must be self-explanatory: insert footnotes for used abbreviations
- Table 3 (title): on a dry weight basis? Please, specify
- Table 3: Species not Specie
- Table 3: P. velutina g/100gDM: g/100g is a percentage as well! Please, clarify (fresh wt./dry wt.) In any case why not uniform these data?
- 433: spp. Instead of sp.
- 434: spp. Instead of sp. Tables must be self-explanatory: (title is too much concise)
- Table 4 (title): which organ? Please, specify
- Table 4: Species not Specie
- Table 4: apply a uniform meas. unit to the entire table
- 502: A study conducted by [111],: Report the author’s name
- Table 5: specify reference organ and units in the title
- Table 5: Species not Specie
- 578: Insert author’s name
- 578-583: too much bla-bla and repetition, consider deleting it and to start the paragraph by 583
- 593: worldly?
- 620: their instead of its
- 621: In terms of…change to: as far as P. is concerned
- 625: these invasive species
- 633: arid and semi-arid areas (or environments).
Author Response
Hi Reviewer and Editor
Thanks for reviewing our manuscript. The following is the response.
Table of Responses- Round 2
We would like to thank the Editor and reviewers for taking time to go through our review article for the second time and providing constructive comments. We have gone through all the comments and our carefully considered responses are appended below. We hope that our responses are satisfactory, however, we stand ready to make further changes should these be required.
Reviewer 3 |
|
Comments/Suggestions |
Responses |
I appreciated your efforts to make the paper easier to read. However, in my opinion, there is still a lot to do in terms of reorganizing the paper and corrections to be made. There are, for instance, numerous repetitions that should be avoided and too much bla-bla that must be eliminated: for example #8.1 and 8.2 could be profitably merged and synthetized. |
Noted. We have gone through all comments and suggestions provided by the reviewers and editors and We hope that our responses are satisfactory, however, we stand ready to make further changes should these be required.
|
Furthermore, there are still some research papers that I’ve already suggested are worthwhile to be included in the bibliography and adequately commented by you |
Noted. Thank you very much for the reminder. We have now included suggested four recent articles (Shackleton et al., 2019, Shackleton and Shackleton, 2018, Shackleton et al., 2017 and Stafford et al., 2017). |
Line: 49: change “universe” with “world” |
Thank you for the suggestion, the word universe has been replaced with the world. |
Line 60: Prosopis |
The word has been properly rewritten as Prosopis |
Line 70: this should be the right place for introducing and summarizing the most relevant and recent review literature (e.g.: The role of invasive alien species in shaping local livelihoods and human well-being: A review, by Ross T. Shackleton, Charlie M. Shackleton and Christian A. Kull, in: Journal of Environmental Management 229 (2019) 145–157). |
Thank you very much. We have cited suggested four recent articles (Shackleton et al., 2019, Shackleton and Shackleton, 2018, Shackleton et al., 2017 and Stafford et al., 2017). |
Line 78: what do you mean by: numerous invasive species association on biodiversity associations have consolidated….?? Please check and rephrase it. |
Thank you and noted. The sentence has been revised, and reads as follows; |
Line 82: Did you mean “extent”? |
Noted, we have corrected it to extent instead of extend |
Line 90: Don’t you think it would be better to first describe Prosopis spp.? (i.e. move #4 to #3) |
Thank you for the kind suggestion. We have moved #3 to #4. |
Line 91: In any case: “The species has invaded several…” change to “The genus has invaded several…” |
Noted, it has been changed and it now reads: “The genus has invaded several…” |
Lines 96-112: The Prosopis species distribution in South Africa is facilitated either by pods that were carried through by flooding events where they would soften in the water and release seeds for germination and softened by water or dispersed by the time when animals consume the ripe seed-pods which they then attempt to digest and excrete the scarified seeds over a dispersed area under the form of scarified seeds [24]. The excreted scarified seeds would then germinate in their new environments. |
Thank you and noted, the sentence has been revised after suggested deletions. |
Lines 114-115 & 116-117: avoid repetition. |
Noted. The sentence reads as follows: With the use of Biome-based procedures when ranking invasive plant species in South Africa [32], Robertson [33] stated that in South Africa, Prosopis plant species were ranked the second species in the Nama-Karoo biome, and the third in the Succulent Karoo biome. |
Lines 132: “crossbreeds; there”: delete semicolon |
Noted and the semicolon has been deleted. |
Lines 188-191: for a clear-cut definition of “invasive”, please, refer to: RICHARDSON D.M., PYŠEK P., REJMÁNEK M, BARBOUR M.G., PANETTA F.D. & WEST C.J., 2000 – Naturalization and invasion of alien plants: concepts and definitions. – Diversity Distrib., 6: 93-107. |
Thank you for the suggestion and well noted and the definition had now been extracted from Richardson et al 2000. |
Lines 191-192: what do you mean by: “…are a vital for biological conservationists,” ? |
Thank you for noting this, the sentence has been rectified and reads as follows: Invasive species are of a concern for biological conservationists |
Lines 201-204: this seems to me an unnecessary repetition |
Noted, the sentence has been deleted. |
Line 208: “in theareas” insert space |
Noted and the space has been inserted |
Lines 210-212: again, unnecessary repetition |
Noted, the sentence has been deleted. |
Lines 221-223: Move to the conclusions |
Thank you for noting this the sentence(s) has been moved to the conclusion and the conclusion was revised to accommodate new sentence. |
Lines 225-228: Move to #5 |
Noted, the sentence has been moved to #5. |
Lines 238-331: Merge and synthetize in only one chapter |
Noted with thanks, it has been merged and synthesized in one chapter/paragraph. |
Lines 248: (citation): use the form 71-76 |
Noted with thanks, it has been rectified. |
Lines 296-299: too much bla-bla…. |
Noted, sentence(s) has been deleted |
Lines 333-334: Unclear sentence: please, rephrase it |
Noted, it has been rephrased. |
Line 334: as for 248: (94-96) |
Noted. |
Line 432: Velutina not Vetulina! |
Thank you and noted, we have replaced the correct spelling of Velutina |
Line 432: avoid repeating g/Kg… |
Noted, and it has been corrected. |
Table 1 & 3 Pallida not Palida! |
Noted, we have used the correct spelling of Pallida. |
Table 2: tables must be self-explanatory: insert footnotes for used abbreviations |
Noted, it has been corrected. The footnote has been inserted in the table |
Table 3 (title): on a dry weight basis? Please, specify |
Noted, yes, it is in dry weight basis. |
Table 3: Species not Specie |
Noted, we have replaced it with Species |
Table 3: P. velutina g/100gDM: g/100g is a percentage as well! Please, clarify (fresh wt./dry wt.) In any case why not uniform these data? |
Noted, the use of units has been corrected. These are in dry weight. For uniformity we have decided to convert all the values to ppm. |
Line 433: spp. Instead of sp. |
Thank you and it has been noted and corrected. In most areas, we have now used full name “species” |
Line 434: spp. Instead of sp. Tables must be self-explanatory: (title is too much concise) |
Thank you for noting this. In most areas, we have now used full name “species” |
Table 4 (title): which organ? Please, specify |
Thank you, it has been specified. Plant parts as leaves and pods |
Table 4: Species not Specie |
Noted, and the word has been rectified to Species. |
Table 4: apply a uniform meas. unit to the entire table |
Thank you for noting this, but we found it difficult to convert some of the units, e.g., g/16g N on Table 4. However, we are ready to make changes if the reviewers have any knowledge on how to convert that into g/100g. |
Line 502: A study conducted by [111],: Report the author’s name |
Noted with thanks the author’s name has been added. |
Table 5: specify reference organ and units in the title |
Noted and it has be done. |
Table 5: Species not Specie |
Noted and rectified as species. |
Line 578: Insert author’s name |
We deleted the sentence together with its citation following the comment below |
Lines 578-583: too much bla-bla and repetition, consider deleting it and to start the paragraph by 583 |
Noted and the sentence has been deleted |
Line 593: worldly? |
Noted, it has been change to world |
Line 620: their instead of its |
Noted, the word has been replaced with ‘their’ |
Line 621: In terms of…change to: as far as P. is concerned |
Noted, this has been changed. |
Line 625: these invasive species |
Noted, and it has been rectified. |
Line 633: arid and semi-arid areas (or environments). |
Noted with thanks, we have corrected the sentence by adding the word environments. |
Table of Responses- Round 2
We would like to thank the Editor and reviewers for taking time to go through our review article for the second time and providing constructive comments. We have gone through all the comments and our carefully considered responses are appended below. We hope that our responses are satisfactory, however, we stand ready to make further changes should these be required.
Reviewer 3 |
|
Comments/Suggestions |
Responses |
I appreciated your efforts to make the paper easier to read. However, in my opinion, there is still a lot to do in terms of reorganizing the paper and corrections to be made. There are, for instance, numerous repetitions that should be avoided and too much bla-bla that must be eliminated: for example #8.1 and 8.2 could be profitably merged and synthetized. |
Noted. We have gone through all comments and suggestions provided by the reviewers and editors and We hope that our responses are satisfactory, however, we stand ready to make further changes should these be required.
|
Furthermore, there are still some research papers that I’ve already suggested are worthwhile to be included in the bibliography and adequately commented by you |
Noted. Thank you very much for the reminder. We have now included suggested four recent articles (Shackleton et al., 2019, Shackleton and Shackleton, 2018, Shackleton et al., 2017 and Stafford et al., 2017). |
Line: 49: change “universe” with “world” |
Thank you for the suggestion, the word universe has been replaced with the world. |
Line 60: Prosopis |
The word has been properly rewritten as Prosopis |
Line 70: this should be the right place for introducing and summarizing the most relevant and recent review literature (e.g.: The role of invasive alien species in shaping local livelihoods and human well-being: A review, by Ross T. Shackleton, Charlie M. Shackleton and Christian A. Kull, in: Journal of Environmental Management 229 (2019) 145–157). |
Thank you very much. We have cited suggested four recent articles (Shackleton et al., 2019, Shackleton and Shackleton, 2018, Shackleton et al., 2017 and Stafford et al., 2017). |
Line 78: what do you mean by: numerous invasive species association on biodiversity associations have consolidated….?? Please check and rephrase it. |
Thank you and noted. The sentence has been revised, and reads as follows; |
Line 82: Did you mean “extent”? |
Noted, we have corrected it to extent instead of extend |
Line 90: Don’t you think it would be better to first describe Prosopis spp.? (i.e. move #4 to #3) |
Thank you for the kind suggestion. We have moved #3 to #4. |
Line 91: In any case: “The species has invaded several…” change to “The genus has invaded several…” |
Noted, it has been changed and it now reads: “The genus has invaded several…” |
Lines 96-112: The Prosopis species distribution in South Africa is facilitated either by pods that were carried through by flooding events where they would soften in the water and release seeds for germination and softened by water or dispersed by the time when animals consume the ripe seed-pods which they then attempt to digest and excrete the scarified seeds over a dispersed area under the form of scarified seeds [24]. The excreted scarified seeds would then germinate in their new environments. |
Thank you and noted, the sentence has been revised after suggested deletions. |
Lines 114-115 & 116-117: avoid repetition. |
Noted. The sentence reads as follows: With the use of Biome-based procedures when ranking invasive plant species in South Africa [32], Robertson [33] stated that in South Africa, Prosopis plant species were ranked the second species in the Nama-Karoo biome, and the third in the Succulent Karoo biome. |
Lines 132: “crossbreeds; there”: delete semicolon |
Noted and the semicolon has been deleted. |
Lines 188-191: for a clear-cut definition of “invasive”, please, refer to: RICHARDSON D.M., PYŠEK P., REJMÁNEK M, BARBOUR M.G., PANETTA F.D. & WEST C.J., 2000 – Naturalization and invasion of alien plants: concepts and definitions. – Diversity Distrib., 6: 93-107. |
Thank you for the suggestion and well noted and the definition had now been extracted from Richardson et al 2000. |
Lines 191-192: what do you mean by: “…are a vital for biological conservationists,” ? |
Thank you for noting this, the sentence has been rectified and reads as follows: Invasive species are of a concern for biological conservationists |
Lines 201-204: this seems to me an unnecessary repetition |
Noted, the sentence has been deleted. |
Line 208: “in theareas” insert space |
Noted and the space has been inserted |
Lines 210-212: again, unnecessary repetition |
Noted, the sentence has been deleted. |
Lines 221-223: Move to the conclusions |
Thank you for noting this the sentence(s) has been moved to the conclusion and the conclusion was revised to accommodate new sentence. |
Lines 225-228: Move to #5 |
Noted, the sentence has been moved to #5. |
Lines 238-331: Merge and synthetize in only one chapter |
Noted with thanks, it has been merged and synthesized in one chapter/paragraph. |
Lines 248: (citation): use the form 71-76 |
Noted with thanks, it has been rectified. |
Lines 296-299: too much bla-bla…. |
Noted, sentence(s) has been deleted |
Lines 333-334: Unclear sentence: please, rephrase it |
Noted, it has been rephrased. |
Line 334: as for 248: (94-96) |
Noted. |
Line 432: Velutina not Vetulina! |
Thank you and noted, we have replaced the correct spelling of Velutina |
Line 432: avoid repeating g/Kg… |
Noted, and it has been corrected. |
Table 1 & 3 Pallida not Palida! |
Noted, we have used the correct spelling of Pallida. |
Table 2: tables must be self-explanatory: insert footnotes for used abbreviations |
Noted, it has been corrected. The footnote has been inserted in the table |
Table 3 (title): on a dry weight basis? Please, specify |
Noted, yes, it is in dry weight basis. |
Table 3: Species not Specie |
Noted, we have replaced it with Species |
Table 3: P. velutina g/100gDM: g/100g is a percentage as well! Please, clarify (fresh wt./dry wt.) In any case why not uniform these data? |
Noted, the use of units has been corrected. These are in dry weight. For uniformity we have decided to convert all the values to ppm. |
Line 433: spp. Instead of sp. |
Thank you and it has been noted and corrected. In most areas, we have now used full name “species” |
Line 434: spp. Instead of sp. Tables must be self-explanatory: (title is too much concise) |
Thank you for noting this. In most areas, we have now used full name “species” |
Table 4 (title): which organ? Please, specify |
Thank you, it has been specified. Plant parts as leaves and pods |
Table 4: Species not Specie |
Noted, and the word has been rectified to Species. |
Table 4: apply a uniform meas. unit to the entire table |
Thank you for noting this, but we found it difficult to convert some of the units, e.g., g/16g N on Table 4. However, we are ready to make changes if the reviewers have any knowledge on how to convert that into g/100g. |
Line 502: A study conducted by [111],: Report the author’s name |
Noted with thanks the author’s name has been added. |
Table 5: specify reference organ and units in the title |
Noted and it has be done. |
Table 5: Species not Specie |
Noted and rectified as species. |
Line 578: Insert author’s name |
We deleted the sentence together with its citation following the comment below |
Lines 578-583: too much bla-bla and repetition, consider deleting it and to start the paragraph by 583 |
Noted and the sentence has been deleted |
Line 593: worldly? |
Noted, it has been change to world |
Line 620: their instead of its |
Noted, the word has been replaced with ‘their’ |
Line 621: In terms of…change to: as far as P. is concerned |
Noted, this has been changed. |
Line 625: these invasive species |
Noted, and it has been rectified. |
Line 633: arid and semi-arid areas (or environments). |
Noted with thanks, we have corrected the sentence by adding the word environments. |
Regards
Hawu and co-authors

Round 3
Reviewer 2 Report
Dear Authors,
no more special comments. Please check carefully the list of references.
Finally, please consider, as already suggested, to adopt table titles more detailed and less concise (i.e. fully explanatory)
Greetings